# ROTATIONOUT AS A REGULARIZATION METHOD FOR NEURAL NETWORK

## ABSTRACT

In this paper, we propose a novel regularization method, RotationOut, for neural networks. Different from Dropout that handles each neuron/channel independently, RotationOut regards its input layer as an entire vector and introduces regularization by randomly rotating the vector. RotationOut can also be used in convolutional layers and recurrent layers with small modifications. We further use a noise analysis method to interpret the difference between RotationOut and Dropout in co-adaptation reduction. Using this method, we also show how to use RotationOut/Dropout together with Batch Normalization. Extensive experiments in vision and language tasks are conducted to show the effectiveness of the proposed method. Codes are available at `https://github.com/RotationOut/RotationOut`.

## 1 INTRODUCTION

Dropout (Srivastava et al., 2014) has proven to be effective for preventing overfitting over many deep learning areas, such as image classification (Shrivastava et al., 2017), natural language processing (Hu et al., 2016) and speech recognition (Amodei et al., 2016). In the years since, a wide range of variants have been proposed for wider scenarios, and most related work focus on the improvement of Dropout structures, i.e., how to drop. For example, drop connect (Wan et al., 2013) drops the weights instead of neurons, evolutional dropout (Li et al., 2016) computes the adaptive dropping probabilities on-the-fly, max-pooling dropout (Wu & Gu, 2015) drops neurons in the max-pooling kernel so smaller feature values have some probabilities to to affect the activations.

These Dropout-like methods process each neuron/channel in one layer independently and introduce randomness by dropping. These architectures are certainly simple and effective. However, randomly dropping independently is not the only method to introduce randomness. Hinton et al. (2012) argues that overfitting can be reduced by preventing co-adaptation between feature detectors. Thus it is helpful to consider other neurons' information when adding noise to one neuron. For example, lateral inhibition noise could be more effective than independent noise.

In this paper, we propose RotationOut as a regularization method for neural networks. RotationOut regards the neurons in one layer as a vector and introduces noise by randomly rotating the vector. Specifically, consider a fully-connected layer with $n$ neurons: $x \in \mathbb{R}^n$. If applying RotationOut to this layer, the output is $\mathcal{R}x$ where $\mathcal{R} \in \mathbb{R}^{n \times n}$ is a random rotation matrix. It rotates the input with random angles and directions, bringing noise to the input. The noise added to a neuron comes not only from itself, but also from other neurons. It is the major difference between RotationOut and Dropout-like methods. We further show that RotationOut uses the activations of the other neurons as the noise to one neuron so that the co-adaptation between neurons can be reduced.

RotationOut uses random rotation matrices instead of unrestricted matrices because the directions of feature vectors are important. Random rotation provides noise to the directions directly. Most neural networks use dot product between the feature vector and weight vector as the output. The network actually learns the direction of the weights, especially when there is a normalization layer (e.g. Batch Normalization (Ioffe & Szegedy, 2015) or Weight Normalization (Salimans & Kingma, 2016)) after the weight layer. Random rotation of feature vecoters introduces noise into the angle between the feature and the weight, making the learning of weights directions more stable. Sabour et al. (2017) also uses the orientation of feature vectors to represent the instantiation parameters in capsules. Another motivation for rotating feature vectors comes from network dissection. Bau et al.

(2017) finds that random rotations of a learned representation can destroy the interpretability which is axis-aligned. Thus random rotating the feature during training makes the network more robust. Even small rotations can be a strong regularization.

We study how RotationOut helps prevent neural networks from overfitting. Hinton et al. (2012) introduces *co-adaptation* to interpret Dropout but few literature give a clear concept of *co-adaptation*. In this paper, we provide a metric to approximate co-adaptations and derive a general formula for noise analysis. Using the formula, we prove that RotationOut can reduce co-adaptations more effectively than Dropout and show how to combine Dropout and Batch Normalization together.

In our experiments, RotationOut can achieve results on par with or better than Dropout and Dropout-like methods among several deep learning tasks. Applying RotationOut after convolutional layers and fully connected layers improves image classification accuracy of ConvNet on CIFAR100 and ImageNet datasets. On COCO datasets, RotationOut also improves the generalization of object detection models. For LSTM models, RotationOut can achieve competitive results with existing RNN dropout method for speech recognition task on Wall Street Journal (WSJ) corpus.

The main contributions of this paper are as follows: We propose RotationOut as a regularization method for neural networks which is different from existing Dropout-like methods that operate on each neuron independently. RotationOut randomly rotates the feature vector and introduces noise to one neuron with other neurons' information. We present a theoretical analysis method for general formula of noise. Using the method, we answer two questions: 1) how noise-based regularization methods reduce co-adaptions and 2) how to combine noise-based regularization methods with Batch Normalization. Experiments in vision and language tasks are conducted to show the effectiveness of the proposed RotationOut method.

**Related Work**  Dropout is effective for fully connected layers. When applied to convolution layers, it is less effective. Ghiasi et al. (2018) argues that information about the input can still be sent to the next layer even with dropout, which causes the networks to overfit (Ghiasi et al., 2018). SpatialDropout (Tompson et al., 2015) drops the entire channel from the feature map. Shake-shake regularization (Gastaldi, 2017) drops the residual branches. Cutout (DeVries & Taylor, 2017) and Dropblock (Ghiasi et al., 2018) drop a continuois square region from the inputs/feature maps.

Applying standard dropout to recurrent layers also results in poor performance (Zaremba et al., 2014; Labach et al., 2019), since the noise caused by dropout at each time step prevents the network from retaining long-term memory. Gal & Ghahramani (2016); Moon et al. (2015); Merity et al. (2017) generate a dropout mask for each input sequence, and keep it the same at every time step so that memory can be retained.

Batch Normalization (BN) (Ioffe & Szegedy, 2015) accelerates deep network training. It is also a regularization to the network, and discourage the strength of dropout to prevent overfitting (Ioffe & Szegedy, 2015). Many modern ConvNet architectures such as ResNet (He et al., 2016) and DenseNet (Huang et al., 2017) do not apply dropout in convolutions. Li et al. (2019) is the first to argue that it is caused by the a variance shift. In this paper, we use the noise analysis method to further explore this problem.

There is a lot of work studying rotations in networks. Rotations on the images (Lenc & Vedaldi, 2015; Simard et al., 2003) are important data augmentation methods. There are also studies about rotation equivalence. Worrall et al. (2017) uses an enriched feature map explicitly capturing the underlying orientations. Marcos et al. (2017) applies multiple rotated versions of each filter to the input to solve problems requiring different responses with respect to the inputs' rotation. The motivations of these work are different from ours. The most related work is network dissection (Bau et al., 2017). They discuss the impact on the interpretability of random rotations of learned features, showing that rotation in training can be a strong regularization.

## 2 ROTATIONOUT

In this section, we first introduce the formulation of RotationOut. Next, we use linear models to demonstrate how RotationOut helps for regularization. In the last part, we discuss the implementation of RotationOut in neural networks.

## 2.1 RANDOM ROTATION MATRIX

A rotation in $D$ dimension is represented by the product between a rotation matrix $\mathcal{R} \in \mathbb{R}^{D \times D}$ and the feature vector $x \in \mathbb{R}^n$. The complexity for random rotation matrix generation and the matrix multiplication are both $O(D^2)$, which would be less efficient than Dropout with $O(D)$ complexity. We consider a special case that uses Givens rotations (Anderson, 2000) to construct random rotation matrices to reduce the complexity.

Let $D = 2d$ be an even number, and $P = [n_1, n_2, \cdots, n_{2d}]$ be a permutation of $\{1, 2, \cdots, D\}$. A rotation matrix can be generated by function $\boldsymbol{M}(\theta, P) = \{r_{ij}\} \in \mathbb{R}^{D \times D}$:

$$r_{ij} = \begin{cases} \cos\theta & \text{if } i = j \\ \sin\theta & \text{if } i = P_l, j = P_{l+d} \\ -\sin\theta & \text{if } i = P_{l+d}, j = P_l \\ 0 & \text{otherwise.} \end{cases} \tag{1}$$

Here $P_l$ represents the $l^{\text{th}}$ element of $P$ where $1 \leq l \leq d$. See Appendix A.1 for some examples of such rotation matrices. Suppose we sample the angle $\theta$ from zero-centered distributions, e.g., truncated Gaussian distribution or uniform distribution and sample the permutation $P$ from $\mathcal{P}$, the set of all permutations of $\{1, 2, \cdots, D\}$, with equal probability. The RotationOut operator $\mathcal{R}$ can be generated using the function $\boldsymbol{M}(P, \theta)$:

$$P \sim \mathcal{P}, \ \theta \sim \text{Unif}(-\Theta, \Theta), \ \mathcal{R} = \frac{1}{\cos\theta} \boldsymbol{M}(P, \theta). \tag{2}$$

Here $1/\cos\theta$ is a normalization term and $\mathcal{R}$ is not a rotation matrix strictly speaking. The random operator generated from Equation 2 have some good properties. 1) The noise is zero centered: $\mathbb{E}_{\mathcal{R}}[\mathcal{R}x] = x$. 2) For any vector $x$ and any random permutation $P$, the angle between $x$ and $\mathcal{R}x$ is determined by angle $\theta$: $\langle x, \mathcal{R}x \rangle = \theta$. 3) For fixed angel $\theta$, there exists $D!/d!$ different rotations. 4) The complexity for random rotation matrix generation and the matrix multiplication are both $O(D)$.

Permutation $P$ draws the rotation direction and angel $\theta$ draws the rotation angle. As an analogy, permutation $P$ is similar to the dropout mask widely used in RNN dropout. There exists $2^D$ different dropout mask ($2^D \ll D!/d!$ for $D > 8$), thus the diversity of random rotation in Equation 1 is sufficient for network training. Angle $\theta$ is similar to the percentage of dropped neurons in Dropout, and the distribution of $\theta$ controls the regularization strength. (Srivastava et al., 2014) used the multiplier's variance to compare Bernoulli dropout and Gaussian dropout. Following this setting, RotationOut is equivalent to Bernoulli Dropout with the keeping rate $p$ and Gaussian dropout with variance $\sigma^2$ if $(1-p)/p = \sigma^2 = \mathbb{E}_\theta \tan^2\theta$.

Reviewing the formulation of the random rotation matrix, it arranges all $D$ dimensions of the input into $d$ pairs randomly, and rotates the two dimension vectors with angle $\theta$ in each pair. Suppose $u$ and $v$ are two dimensions/neurons in one pair, the outputs of $u$ and $v$ after RotationOut are

$$\begin{bmatrix} u' \\ v' \end{bmatrix} = \begin{bmatrix} 1 & \tan\theta \\ -\tan\theta & 1 \end{bmatrix} \begin{bmatrix} u \\ v \end{bmatrix} = \begin{bmatrix} u + v\tan\theta \\ v - u\tan\theta \end{bmatrix} \tag{3}$$

The noise of $u'$ comes from $v$ and the noise of $v'$ comes from $u$ since $\theta$ is random. Note that the pairs are randomly arranged, thus RotationOut uses all other dimensions/neurons as the noise for one dimension/neuron of the feature vector. With RotationOut, the neurons are trained to work more independently since one neuron has to regard the activation of other neurons as noise. Thus the co-adaptations are reduced.

Consider Gaussian dropout, the outputs are $u' = u + u\epsilon, v' = v + v\epsilon$ where $\mathbb{E}\epsilon = 0, \mathbb{E}\epsilon^2 = \mathbb{E}_\theta \tan^2\theta$. The difference between Gaussian dropout and RotationOut is the source of noise, i.e., the Gaussian dropout noise for one neuron comes from itself while the RotationOut noise comes from other neurons.

## 2.2 ROTATIONOUT IN LINEAR MODELS

First we consider a simple case of applying RotationOut to the classical problem of linear regression. Let $\{(x_i, y_i)\}_{i=1}^N$ be the dataset where $x_i \in \mathbb{R}^D, y_i \in \mathbb{R}$. Linear regression tries to find the weight

$\boldsymbol{w} \in \mathbb{R}^D$ that minimizes $\sum_{i=1}^{N}(y_i - \boldsymbol{w}^{\mathrm{T}}\boldsymbol{x}_i)^2$. When applied RotationOut to each $\boldsymbol{x}_i$, we generate $\mathcal{R}_i$ from Equation 2 for each $\boldsymbol{x}_i$. The objective function becomes:

$$\min_{\boldsymbol{w}} \mathbb{E}_{\mathcal{R}}\Big[\sum_{i=1}^{N}(y_i - \boldsymbol{w}^{\mathrm{T}}\mathcal{R}_i\boldsymbol{x}_i)^2\Big]. \tag{4}$$

Denote $\boldsymbol{y} = [y_1, y_2, \cdots, y_n]^{\mathrm{T}} \in \mathbb{R}^N$, $\boldsymbol{X} = [\boldsymbol{x}_1, \boldsymbol{x}_2, \cdots, \boldsymbol{x}_n]^{\mathrm{T}} \in \mathbb{R}^{N \times D}$. To compare RotationOut with Dropout with keep rate $p$, we suppose $\mathbb{E}_\theta \tan^2 \theta = (1-p)/p = \lambda$. Equation 4 reduces to:

$$\min_{\boldsymbol{w}} \|\boldsymbol{y} - \boldsymbol{X}\boldsymbol{w}\|^2 + \lambda \boldsymbol{w}^{\mathrm{T}} \frac{\mathrm{trace}(\boldsymbol{X}^{\mathrm{T}}\boldsymbol{X})\boldsymbol{I} - \boldsymbol{X}^{\mathrm{T}}\boldsymbol{X}}{D-1}\boldsymbol{w}. \tag{5}$$

Details see Appendix A.2. Solutions to Equation 5 (LR with Rotation) and the mirror problem with dropout (Srivastava et al., 2014) are :

$$\boldsymbol{w}_{\mathrm{Rot}} = \Big[\boldsymbol{X}^{\mathrm{T}}\boldsymbol{X} + \lambda \frac{\mathrm{trace}(\boldsymbol{X}^{\mathrm{T}}\boldsymbol{X})\boldsymbol{I} - \boldsymbol{X}^{\mathrm{T}}\boldsymbol{X}}{D-1}\Big]^{-1}\boldsymbol{X}^{\mathrm{T}}\boldsymbol{y}$$

$$\boldsymbol{w}_{\mathrm{Drop}} = \big[\boldsymbol{X}^{\mathrm{T}}\boldsymbol{X} + \lambda \mathrm{diag}(\boldsymbol{X}^{\mathrm{T}}\boldsymbol{X})\big]^{-1}\boldsymbol{X}^{\mathrm{T}}\boldsymbol{y} \tag{6}$$

Therefore, linear regression with RotationOut and Dropout are equivalent to ridge regression with different regularization terms. Set $\lambda = 1$ (Dropout rate $p = 0.5$) for simplicity. LR with Dropout doubles the diagonal elements of $\boldsymbol{X}^{\mathrm{T}}\boldsymbol{X}$ to make the problem numerical stable. LR with RotationOut is more close to ridge regression:

$$\boldsymbol{X}^{\mathrm{T}}\boldsymbol{X} + \frac{\mathrm{trace}(\boldsymbol{X}^{\mathrm{T}}\boldsymbol{X})\boldsymbol{I} - \boldsymbol{X}^{\mathrm{T}}\boldsymbol{X}}{D-1} = \frac{D-2}{D-1}\Big[\boldsymbol{X}^{\mathrm{T}}\boldsymbol{X} + \frac{\mathrm{trace}(\boldsymbol{X}^{\mathrm{T}}\boldsymbol{X})}{D-2}\boldsymbol{I}\Big] \tag{7}$$

The condition number of Equation 7 and the LR with RotationOut problem is up bounded by $D-1$. For the Dropout case, if some data dimensions have extremely small variances, both $\boldsymbol{X}^{\mathrm{T}}\boldsymbol{X}$ and $\mathrm{diag}(\boldsymbol{X}^{\mathrm{T}}\boldsymbol{X})$ are ill-conditioned. LR with Dropout problem has unbounded condition number.

Next we consider an $m$-way classification model of logistic regression. The input is $\boldsymbol{x} \in \mathbb{R}^D$ and the weights are $\boldsymbol{W} = [\boldsymbol{w}_1, \boldsymbol{w}_2, \cdots, \boldsymbol{w}_m] \in \mathbb{R}^{m \times D}$. The probability that the input belongs to the $k$ category is:

$$p_k = \frac{\exp(\boldsymbol{w}_k \boldsymbol{x})}{\sum_i \exp(\boldsymbol{w}_i \boldsymbol{x})} = \frac{\exp(\|\boldsymbol{w}_k\|\|\boldsymbol{x}\|\cos\theta_k)}{\sum_i \exp(\|\boldsymbol{w}_i\|\|\boldsymbol{x}\|\cos\theta_i)}. \tag{8}$$

In Equation 8, $\theta_i$ denotes the angel between $\boldsymbol{x}$ and $\boldsymbol{w}_i$. Assume that the length of each weights $\boldsymbol{w}_i$ are very close, the input $\boldsymbol{x}$ belongs to the $k$ category if $\boldsymbol{x}$ is most close to $\boldsymbol{w}_k$ in angle.

Consider a hard sample case that $\theta_i < \theta_j$ are the two smallest weight-data angles. But $\theta_i$ and $\theta_j$ are very close: $\theta_i \approx \theta_j$, i.e., the data are close to the decision boundary. The model should classify the data correctly but could make mistakes if there is some noise. Applying RotationOut, the angle between the data and the weights can be changed, and the new angles can be $\widehat{\theta}_i > \widehat{\theta}_j$. To classify the data correctly, there should be a gap between $\theta_i$ and $\theta_j$. In other words, the decision boundary changed from $\theta_i < \theta_j$ to $\theta_i < \theta_j - \Theta$ where $\Theta$ is a positive constant that depends on the regularization. Thus RotationOut can be regarded as a margin-based hard sample mining.

Here we provide an intuitive understanding of how Dropout with low keep rates leads to lower performance. Randomly zeroing units, Dropout method also rotates the feature vector. A lower keep rate results in a bigger rotation angle: $\cos^2\theta = \frac{(\sum_i p_i x_i^2)^2}{\sum_i x_i^2 \sum_i p_i^2 x_i^2} \approx \frac{(\mathbb{E}p_i)^2}{\mathbb{E}p_i^2} = p$. Consider the last hidden layer in neural networks, it is similar to logistic regression on the features. If one feature $\boldsymbol{x}$ is most close to $\boldsymbol{w}_k$, it belongs to the $k^{\mathrm{th}}$. A lower keep rate Dropout would rotate the feature with a bigger angle, and the Dropout output can be most close to another weight with higher probability, which may hurts the training.

## 2.3 ROTATIONOUT IN NEURAL NETWORKS

Consider a neural network with $L$ hidden layers. Let $\boldsymbol{x}^l$, $\boldsymbol{y}^l$, and $\boldsymbol{W}^l$ denote the vector of inputs, the vector of output before activation, and the weights for the layer $l$. Let $\mathcal{R}$ be generated from

Equation 2 and $a$ be the activation function, for example Rectified Linear Unit (ReLU). The MLP feed-forward operation with RotationOut in training time can be:

$$\widetilde{\boldsymbol{x}}^l = \mathcal{R}(\boldsymbol{x}^l - \mathbb{E}[\boldsymbol{x}^l]) + \mathbb{E}[\boldsymbol{x}^l], \quad \boldsymbol{y}^l = \boldsymbol{W}^l \widetilde{\boldsymbol{x}}^l, \quad \boldsymbol{x}^{l+1} = a(\boldsymbol{y}^l). \tag{9}$$

We rotate the zero-centered features and then add the expectation back. The reasons will be explained later. Here we give an intuitive understanding. If features are not zero-centered, we do not know the exact regularization strength. Suppose all features elements are in one interval, say $1 < x < 2$. The angle between any two feature vectors is a sharp angle. In this case a rotation angle of $\pi/4$ would be too big. It is the same for Dropout. The regularization strength is influenced by the mean value of features which we may not know. At test time, the RotationOut operation is removed.

Consider 2D case for example, the input for 2D convolutional layers are three dimensional: number of channels $C$, width $H$ and height $W$:

$$\boldsymbol{X} = \{\boldsymbol{x}_{hw}\}, \boldsymbol{x}_{hw} \in \mathbb{R}^C, 1 \leq h \leq H, 1 \leq w \leq W \tag{10}$$

We regard each $\boldsymbol{x}_{hw}$ as a feature vector with semantic information for each position $(h, w)$, and apply rotation to each position. As Ghiasi et al. (2018) argued, the convolutional feature maps are spatially correlated, so information can still flow through convolutional layers if features are dropped out randomly. Similarly, if we rotate feature vectors in different positions with random directions, random directions offset each other and result in no rotation. So we rotate all feature vectors with the same directions but different angles. The operation on convolutional feature maps can be:

$$\begin{aligned} P &\sim \mathcal{P}, \quad \theta_{11}, \cdots, \theta_{HW} \sim \text{Unif}(0, \Theta), \\ \forall h, w \quad \mathcal{R}_{hw} &= \boldsymbol{M}(P, \theta_{hw}), \\ \widetilde{\boldsymbol{x}}_{hw} &= \mathcal{R}_{hw}(\boldsymbol{x}_{hw} - \mathbb{E}\boldsymbol{x}_{hw}) + \mathbb{E}\boldsymbol{x}_{hw}. \end{aligned} \tag{11}$$

The operation for general convolutional networks are very similar. Also note that RotationOut can combined with DropBlock (Ghiasi et al., 2018) easily: only rotating features in a continuous block. Experiments show that the combination can get extra performance gain. As mentioned in Section 3.1, the rotation directions defined by $P$ is similar to the dropout mask in RNN drops. RotationOut can also be used in recurrent networks following Equation 11.

## 2.4 NOISE ANALYSIS

In this section, we first study the general formula of adding noise. Using the formula, we show how introducing randomness/noise helps reduce co-adaptations and why RotationOut is more efficient than the vanilla dropout. Strictly speaking, the co-adaptations describe the dependence between neurons. The mutual information between two neurons may be the best metric to define co-adaptations. To compute mutual information, we need the exact distributions of neurons, which are generally unknown. So we consider the correlation coefficient to evaluate co-adaptations, which only need the first and second moment. Moreover, if we assume the distributions of neurons are Gaussian, correlation coefficient and mutual information are equivalent in co-adaptations evaluation.

Suppose $\boldsymbol{x} \in \mathbb{R}^D$ is the activations of one hidden layer. Let $\mathbb{E}[\boldsymbol{x}] = \boldsymbol{c} \in \mathbb{R}^D$, $\text{Var}[\boldsymbol{x}] = \Sigma \in \mathbb{R}^{D \times D}$. The ideal situation is that $\Sigma = \text{diag}\Sigma$, i.e., the neurons are mutually independent. We define the co-adaptations as the distance between $\Sigma$ and $\Sigma = \text{diag}(\Sigma)$.

$$\text{co}(\boldsymbol{x}) = \frac{\|\Sigma - \text{diag}(\Sigma)\|_1}{\|\text{diag}(\Sigma)\|_1} = \frac{\|\Sigma - \text{diag}(\Sigma)\|_1}{\text{trace}(\Sigma)} \tag{12}$$

Here $\text{trace}(\Sigma)$ is a normalization term that defines the regularization strength. Let $\widetilde{\boldsymbol{x}}$ be the out of $\boldsymbol{x}$ with arbitrary noise (e.g. Dropout or RotationOut). We assume that the noise should follow two assumptions: 1) zero-center: $\mathbb{E}[\widetilde{\boldsymbol{x}}|\boldsymbol{x}] = \boldsymbol{x}$; 2) non-trivial: $\text{Var}[\widetilde{\boldsymbol{x}}|\boldsymbol{x}] \neq \boldsymbol{O}$ (avoid that $\widetilde{\boldsymbol{x}}$ always equals to $\boldsymbol{x}$). Consider the law of total variance, we have:

$$\text{Var}[\widetilde{\boldsymbol{x}}] = \mathbb{E}\left[\text{Var}[\widetilde{\boldsymbol{x}}|\boldsymbol{x}]\right] + \text{Var}\left[\mathbb{E}[\widetilde{\boldsymbol{x}}|\boldsymbol{x}]\right] = \mathbb{E}\left[\text{Var}[\widetilde{\boldsymbol{x}}|\boldsymbol{x}]\right] + \text{Var}[\boldsymbol{x}] \tag{13}$$

Let $\widetilde{\boldsymbol{x}}_{\text{Drop}}$ be the out of $\boldsymbol{x}$ after Dropout with drop rate $p$, and $\widetilde{\boldsymbol{x}}_{\text{Rot}}$ be the out of $\boldsymbol{x}$ after RotationOut with $\mathbb{E}_\theta \tan^2 \theta = (1-p)/p$, we have Lemma 1 (proof see Appendix A.3):

**Lemma 1.** $\text{Var}[\widetilde{\boldsymbol{x}}_{\text{Drop}}|\boldsymbol{x}] = \frac{1-p}{p}\text{diag}(\boldsymbol{x}\boldsymbol{x}^{\text{T}})$, $\text{Var}[\widetilde{\boldsymbol{x}}_{\text{Rot}}|\boldsymbol{x}] = \frac{1-p}{p(D-1)}(\boldsymbol{x}^{\text{T}}\boldsymbol{x}\boldsymbol{I} - \boldsymbol{x}\boldsymbol{x}^{\text{T}})$.

Note that $\mathbb{E}[\boldsymbol{x}\boldsymbol{x}^{\mathrm{T}}] = \Sigma + \boldsymbol{c}\boldsymbol{c}^{\mathrm{T}}, \mathbb{E}[\boldsymbol{x}^{\mathrm{T}}\boldsymbol{x}] = \mathrm{trace}(\Sigma) + \boldsymbol{c}^{\mathrm{T}}\boldsymbol{c}$, we have:

$$\mathrm{Var}[\widetilde{\boldsymbol{x}}_{\mathrm{Drop}}] = \Sigma + \frac{1-p}{p}\mathrm{diag}(\Sigma + \boldsymbol{c}\boldsymbol{c}^{\mathrm{T}})$$

$$\mathrm{Var}[\widetilde{\boldsymbol{x}}_{\mathrm{Rot}}] = \Sigma + \frac{1-p}{p}\frac{\mathrm{trace}(\Sigma)\boldsymbol{I} - \Sigma + \boldsymbol{c}^{\mathrm{T}}\boldsymbol{c}\boldsymbol{I} - \boldsymbol{c}\boldsymbol{c}^{\mathrm{T}}}{D-1}$$

(14)

We can compute the co-adaptations of $\widetilde{\boldsymbol{x}}$ (Assume $\boldsymbol{c} = \boldsymbol{0}$):

$$\mathrm{co}(\widetilde{\boldsymbol{x}}_{\mathrm{Drop}}) = \frac{\|\widetilde{\boldsymbol{x}}_{\mathrm{Drop}} - \mathrm{diag}(\widetilde{\boldsymbol{x}}_{\mathrm{Drop}})\|_1}{\mathrm{trace}(\widetilde{\boldsymbol{x}}_{\mathrm{Drop}})} = \frac{\|\Sigma - \mathrm{trace}(\Sigma)\|_1}{\frac{1}{p}\mathrm{trace}(\Sigma)} = p\,\mathrm{co}(\boldsymbol{x})$$

$$\mathrm{co}(\widetilde{\boldsymbol{x}}_{\mathrm{Rot}}) = \frac{(1 - \frac{1-p}{p(D-1)})\|\Sigma - \mathrm{trace}(\Sigma)\|_1}{\mathrm{trace}(\Sigma) + \frac{1-p}{p}\frac{D\mathrm{trace}(\Sigma)-\mathrm{trace}(\Sigma)}{D-1}} = (p - \frac{1-p}{D-1})\,\mathrm{co}(\boldsymbol{x})$$

(15)

Under zero-center assumption, Dropout with keep rate $p$ reduces co-adaptation by $p$ times, and the equivalent RotationOut reduces co-adaptation by $p - \frac{1-p}{D-1}$ times.

We take a close look at the correlation coefficient to see what makes the difference. Let $\boldsymbol{x}_i$ be the $i^{\mathrm{th}}$ element of $\boldsymbol{x}$. Recall Equation 13, we have:

$$\left|\mathrm{cor}(\widetilde{\boldsymbol{x}}_i, \widetilde{\boldsymbol{x}}_j)\right| = \frac{\left|\mathrm{cov}(\boldsymbol{x}_i, \boldsymbol{x}_j) + \mathbb{E}\left[\mathrm{cov}(\widetilde{\boldsymbol{x}}_i, \widetilde{\boldsymbol{x}}_j|\boldsymbol{x})\right]\right|}{\sqrt{(Var\left[\boldsymbol{x}_i\right] + \mathbb{E}\left[Var[\widetilde{\boldsymbol{x}}_i|\boldsymbol{x}]\right])(Var\left[\boldsymbol{x}_j\right] + \mathbb{E}\left[Var[\widetilde{\boldsymbol{x}}_j|\boldsymbol{x}]\right])}}$$

(16)

For Dropout-and other dropout-like methods, they add noise to different neurons independently, so $\mathrm{cov}(\widetilde{\boldsymbol{x}}_i, \widetilde{\boldsymbol{x}}_j|\boldsymbol{x}) = 0$. The only term to reduce correlation coefficients in Equation 16 is $\mathbb{E}\left[Var[\widetilde{\boldsymbol{x}}_j|\boldsymbol{x}]\right]$. Under out non-trivial noise assumption, $Var[\widetilde{\boldsymbol{x}}_j|\boldsymbol{x}]$ is always positive. Thus non-trivial noise can always reduce co-adaptations. For RotationOut, there is another term to reduce correlation coefficients: $\mathbb{E}\left[\mathrm{cov}(\widetilde{\boldsymbol{x}}_i, \widetilde{\boldsymbol{x}}_j|\boldsymbol{x})\right] = -\frac{1-p}{p(D-1)}\mathrm{cov}(\boldsymbol{x}_i, \boldsymbol{x}_j)$ and typically $0 < \frac{1-p}{p(D-1)} < 1$. In addition to increasing the uncertainty of each neuron as Dropout does, RotationOut can also reduce the correlation between two neurons. In other words, *inhibition noise*.

Here we explain why we need a zero-center assumption and rotate the zero-centered features in Section 2.3. Equation 14 and 16 show that the non-zero mean value can further reduce the co-adaptations. If we do not know the exact mean value, we do not know the exact regularization strength. Suppose the neurons $x \sim \mathcal{N}(0,1)$ follow a normal distribution, and we apply Dropout on the ReLU activations $y = \mathrm{ReLU}(x)$. With a keep rate 0.9, Dropout reduces the co-adaptations by 0.86 times, while Dropout reduces the co-adaptations by 0.61 times with a keep rate 0.7, which is a non-linear mapping and influenced by the mean value. We rotate/drop the zero-centered features so that the regularization strength is independent with the mean value.

## 3 EXPERIMENTS

In this section, we evaluate the performance of RotationOut for image classification, object detection, and speech recognition. First, we conduct detailed ablation studies with CIFAR100 dataset. Next, we compare RotationOut with other regularization techniques using more data and higher resolution. We test on two tasks: image classification on ILSVRC dataset and object detection on COCO dataset.

### 3.1 ABLATION STUDY ON CIFAR100

The CIFAR100 dataset consists of 60,000 colour images of size $32 \times 32$ pixels and 100 classes. The official version of the dataset is split into a training set with 50,000 images and a test set with 10,000 images. We conduct image classification experiments on the dataset.

Our focus is on the regularization abilities, so the experiment settings for different regularization techniques are the same. We follow the setting from He et al. (2016). The network inputs are $32 \times 32$ and normalized using per-channel mean and standard deviation. The data augmentation methods are as follows: first zero-pad the images with 4 pixels on each side to obtain a $40 \times 40$ pixel image, then randomly crop a $32 \times 32$ pixel image, and finally mirror the images horizontally with

50% probability. For all of these experiments, we use the same optimizer: training for 64k iterations with batches of 128 images using SGD, momentum of 0.9, and weight decay of 1e-5. We start with a learning rate of 0.1, divide it by 10 at 32k and 48k iterations, and terminate training at 64k iterations. For each run, we record the best validation accuracy and the avergae validation accuracy of the last 10 epochs. Each experiment is repeated 5 times and we report the top 1 (best and avergae) validation accuracy as "mean $\pm$ standard deviation" of the 5 runs.

We compare the regularization abilities of RotationOut and Dropout on two classical architectures: ResNet110 from He et al. (2016) and WideResNet28-10 from Zagoruyko & Komodakis (2016). ResNet110 is a deep but not so wide architecture using $18 \times 3$ BasicBlocks (Zagoruyko & Komodakis, 2016) in three residual stages. The feature map sizes are $\{32, 16, 8\}$ respectively and the numbers of filters are $\{16, 32, 64\}$ respectively. WideResNet28-10 is a wide but not so deep architecture using $4 \times 3$ BasicBlocks in three residual stages. The feature map sizes are $\{32, 16, 8\}$ respectively and the numbers of filters are $\{160, 320, 640\}$ respectively. For ResNet110, we only apply RotationOut or Dropout (with the same rate) to all convolutional layers in the third residual stages. FOr WideResNet28-10, we apply RotationOut or Dropout (with the same keep rate) to all convolutional layers in the second and third residual stages since WideResNet28-10 has much more parameters.

As mentioned ealier, we can use different distributions to generate $\theta$. and the regularization strength is controlled by $\mathbb{E} \tan \theta^2 = 1/p - 1$. We compare RotationOut with the corresponding Dropout. We tried different distributions and found that the performance difference is very small. We report the results of Gaussian distributions here.

Table 1: Top 1 accuracy of Dropout and corresponding RotationOut on CIFAR100

(a) ResNet110: Standard Dropout

| keep rate | Avg top-1(%) | Best top-1(%) |
|---|---|---|
| 0 | $71.93 \pm 0.16$ | $72.12 \pm 0.18$ |
| 0.9 | $73.13 \pm 0.30$ | $73.32 \pm 0.28$ |
| 0.8 | $\mathbf{73.33} \pm 0.27$ | $\mathbf{73.59} \pm 0.28$ |
| 0.7 | $72.42 \pm 0.23$ | $72.71 \pm 0.14$ |
| 0.6 | $71.79 \pm 0.20$ | $72.10 \pm 0.21$ |

(b) ResNet110: $\tan \theta \sim \mathcal{N}(0, \sigma^2)$

| $\sigma$ | Avg top-1(%) | Best top-1(%) |
|---|---|---|
| 0 | $71.93 \pm 0.16$ | $72.12 \pm 0.18$ |
| 0.333 | $74.23 \pm 0.25$ | $74.41 \pm 0.16$ |
| 0.500 | $\mathbf{74.14} \pm 0.11$ | $\mathbf{74.35} \pm 0.13$ |
| 0.655 | $73.13 \pm 0.14$ | $73.45 \pm 0.11$ |
| 0.816 | $71.83 \pm 0.35$ | $72.17 \pm 0.33$ |

(c) WideResNet28: Standard Dropout

| keep rate | Avg top-1(%) | Best top-1(%) |
|---|---|---|
| 0 | $78.05 \pm 0.23$ | $78.20 \pm 0.21$ |
| 0.9 | $78.61 \pm 0.09$ | $78.78 \pm 0.10$ |
| 0.8 | $\mathbf{78.77} \pm 0.20$ | $\mathbf{78.91} \pm 0.19$ |
| 0.7 | $78.75 \pm 0.15$ | $78.87 \pm 0.13$ |
| 0.6 | $78.55 \pm 0.07$ | $78.75 \pm 0.18$ |

(d) WideResNet28: $\tan \theta \sim \mathcal{N}(0, \sigma^2)$

| $\sigma$ | Avg top-1(%) | Best top-1(%) |
|---|---|---|
| 0 | $78.05 \pm 0.23$ | $78.20 \pm 0.21$ |
| 0.333 | $78.94 \pm 0.22$ | $79.09 \pm 0.22$ |
| 0.500 | $79.47 \pm 0.14$ | $79.60 \pm 0.12$ |
| 0.655 | $79.69 \pm 0.11$ | $78.80 \pm 0.15$ |
| 0.816 | $\mathbf{79.76} \pm 0.32$ | $\mathbf{79.93} \pm 0.33$ |

Table 1 shows the results on CIFAR100 dataset with two architectures. Table 1a and 1b are the results for ResNet110. Table 1c and 1d are the results for WideResNet28-10. Results in the same row compare the regularization abilities of Dropout and the equivalent keep rate RotationOut. We can find dropping too many neurons is less effective and may hurt training. Since WideResNet28-10 has much more parameters, the best performance is from a heavier regularization.

## 3.2 Experiments with more data and higher resolution

**ImageNet Classification.** The ILSVRC 2012 classification dataset contains 1.2 million training images and 50,000 validation images with 1,000 categories. We following the training and test schema as in (Szegedy et al., 2015; He et al., 2016) but train the model for 240 epochs. The learning rate is decayed by the factor of 0.1 at 120, 190 and 230 epochs. We apply RotationOut with with normal distribution of tangent $\mathbb{E} \tan \theta^2 = 1/4$ to convolutional layers in Res3 and Res4 as well as the last fully connected layer. As mentioned earlier, RotationOut is easily combined with DropBlock idea. We rotate features in a continuous block size of $7 \times 7$ in Res3 and $3 \times 3$ in Res4.

Table 2 shows the results of some state of the art methods and our results. Our results are average over 5 runs. Results of other methods are from Ghiasi et al. (2018), and also regularize on Res3 and Res4. Our result is significantly better than Dropout and SpatialDropout. By using the DropBlock idea, RotationOut can get competitive results compared with state of the art methods and get a 2.07% improvement compared with the baseline.

Table 2: Comparison with state of the art: Top 1 accuacy of ResNet50 on ImageNet Validation

| Model | top-1(%) | top-5(%) |
|---|---|---|
| ResNet-50 (He et al., 2016) | $76.51 \pm 0.07$ | $93.20 \pm 0.05$ |
| ResNet-50 + dropout(kp=0.7)(Srivastava et al., 2014) | $76.80 \pm 0.04$ | $93.41 \pm 0.04$ |
| ResNet-50 + DropPath(kp=0.9)(Larsson et al., 2016) | $77.10 \pm 0.08$ | $93.50 \pm 0.05$ |
| ResNet-50 + SpatialDropout(kp=0.9) (Tompson et al., 2015) | $77.41 \pm 0.04$ | $93.74 \pm 0.02$ |
| ResNet-50 + Cutout (DeVries & Taylor, 2017) | $76.52 \pm 0.07$ | $93.21 \pm 0.04$ |
| ResNet-50 + DropBlock(kp=0.9) (Ghiasi et al., 2018) | $78.13 \pm 0.05$ | $94.02 \pm 0.02$ |
| ResNet-50 + RotationOut | $77.87 \pm 0.34$ | $93.94 \pm 0.17$ |
| ResNet-50 + RotationOut (Block) | $\mathbf{78.58} \pm 0.48$ | $94.27 \pm 0.24$ |

**COCO Object Detection.** Our proposed method can also be used in other vision tasks, for example Object Detection on MS COCO (Lin et al., 2014). In this task, we use RetinaNet (Lin et al., 2017) as the detection method and apply RotationOut to the ResNet backbone. We use the same hyperparameters as in ImageNet classification. We follow the implementation details in (Ghiasi et al., 2018): resize images between scales [512, 768] and then crop the image to max dimension 640. The model are initialized with ImageNet pretraining and trained for 35 epochs with learning decay at 20 and 28 epochs. We set $\alpha = 0.25$ and $\gamma = 1.5$ for focal loss, a weight decay of 0.0001, a momentum of 0.9 and a batch size of 64. The model is trained on COCO train2017 and evaluated on COCO val2017. We compare our result with DropBlock (Ghiasi et al., 2018) as table 3 shows.

Table 3: Object detection in COCO using RetinaNet and ResNet-50 FPN backbone

| Model | Initialization | AP | AP50 | AP75 |
|---|---|---|---|---|
| RetinaNet | ImageNet | 36.5 | 55.0 | 39.1 |
| RetinaNet, no DropBlock | Random | 36.8 | 54.6 | 39.4 |
| RetinaNet, Dropout, keep_prob $= 0.9$ | Random | 37.9 | 56.1 | 40.6 |
| RetinaNet, keep_prob $= 0.9$, block_size $= 5$ | Random | 38.4 | 56.4 | 41.2 |
| RetinaNet, RotationOut | ImageNet | 38.2 | 56.2 | 41.0 |
| RetinaNet, RotationOut (Block) | ImageNet | **38.7** | 56.6 | 41.4 |

Due to limited computing resources, we finetune the model from PyTorch library's pretraining ImageNet classification models while DropBlock method trained the model from scratch. We think it is fair to compare DropBlock method since the initialization does not help increase the results as showed in the first two rows. Our RotationOut can still have additional 0.3 AP based on the DropBlock result.

## 4 CONCLUSION

In this work, we introduce RotationOut as an alternative for dropout for neural network. RotationOut adds continuous noise to data/features and keep the semantics. We further establish an analysis of noise to show how co-adaptations are reduced in neural network and why dropout is more effective than dropout. Our experiments show that applying RotationOut in neural network helps training and increase the accuracy. Possible direction for further work is the theoretical analysis of co-adaptations. As discussed earlier, the proposed correlation analysis is not optimal. It cannot explain the difference between standard Dropout and Gaussian dropout. Also it can not ex-

plain some methods such as Shake-shake regularization. Further work on co-adaptation analysis can help better understand noise-based regularization methods.

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

# A APPENDIX

## A.1 RANDOM ROTATION MATRIX

One example of such a matrix that rotates the $(1, 3)$ dimensions and $(2, 4)$ dimensions can be:

$$\boldsymbol{M}(\theta, [3, 2, 1, 4]) = \begin{bmatrix} \cos\theta & 0 & -\sin\theta & 0 \\ 0 & \cos\theta & 0 & \sin\theta \\ \sin\theta & 0 & \cos\theta & 0 \\ 0 & -\sin\theta & 0 & \cos\theta \end{bmatrix}. \tag{17}$$

In Section 2, we mentioned the complexity of RotationOut is $O(D)$. It is because we can avoid matrix multiplications to get $\mathcal{R}\boldsymbol{x}$. For example, let the $\mathcal{R}$ be the operator generated by Equation 17, we have:

$$\mathcal{R}\boldsymbol{x} = \boldsymbol{x} + \tan\theta \begin{bmatrix} 0 & 0 & -1 & 0 \\ 0 & 0 & 0 & 1 \\ 1 & 0 & 0 & 0 \\ 0 & -1 & 0 & 0 \end{bmatrix} \boldsymbol{x}. \tag{18}$$

The sparse matrix in Equation 18 is similar to a combine of permutation matrix, and we do not need matrix multiplications to get the output. The output can be get by slicing and an elementwise multiplication: $\boldsymbol{x}[3, 4, 1, 2] * [-1, 1, 1, -1]$.

## A.2 MARGINALIZING LINEAR REGRESSION

Recall that $\mathbb{E}_{\mathcal{R}} = \boldsymbol{I}$, the marginalizing linear regression expression:

$$\mathbb{E}_{\mathcal{R}}\Big[\sum_{i=1}^{N}(y_i - \boldsymbol{w}^{\mathrm{T}}\mathcal{R}_i\boldsymbol{x}_i)^2\Big]$$

$$=\mathbb{E}_{\mathcal{R}}\Big[\sum_{i=1}^{N}(y_i - \boldsymbol{w}^{\mathrm{T}}\boldsymbol{x}_i + \boldsymbol{w}^{\mathrm{T}}(\boldsymbol{I} - \mathcal{R}_i)\boldsymbol{x}_i)^2\Big]$$

$$=\Big[\sum_{i=1}^{N}(y_i - \boldsymbol{w}^{\mathrm{T}}\boldsymbol{x}_i)^2\Big] + \sum_{i=1}^{N}(y_i - \boldsymbol{w}^{\mathrm{T}}\boldsymbol{x}_i)\mathbb{E}_{\mathcal{R}}\Big[\boldsymbol{w}^{\mathrm{T}}(\boldsymbol{I} - \mathcal{R}_i)\boldsymbol{x}_i\Big] + \boldsymbol{w}^{\mathrm{T}}\mathbb{E}_{\mathcal{R}}\Big[(\boldsymbol{I} - \mathcal{R}_i)\boldsymbol{x}_i\boldsymbol{x}_i^{\mathrm{T}}(\boldsymbol{I} - \mathcal{R}_i)^{\mathrm{T}}\Big]\boldsymbol{w}$$

$$=\|\boldsymbol{y} - \boldsymbol{X}\boldsymbol{w}\|^2 + \boldsymbol{w}^{\mathrm{T}}\sum_{i=1}^{N}\mathrm{Var}_{\mathcal{R}}[(\boldsymbol{I} - \mathcal{R}_i)\boldsymbol{x}_i]\boldsymbol{w}$$

$$=\|\boldsymbol{y} - \boldsymbol{X}\boldsymbol{w}\|^2 + \boldsymbol{w}^{\mathrm{T}}\sum_{i=1}^{N}\mathrm{Var}_{\mathcal{R}}[\mathcal{R}_i\boldsymbol{x}_i]\boldsymbol{w}$$

$$\tag{19}$$

From Lemma one, we have $\mathrm{Var}_{\mathcal{R}}[\mathcal{R}_i\boldsymbol{x}_i] = \frac{1-p}{pD-p}(\boldsymbol{x}_i^{\mathrm{T}}\boldsymbol{x}_i\boldsymbol{I} - \boldsymbol{x}_i^{\mathrm{T}}\boldsymbol{x}_i)$. Write the second term of Equation 19 in the matrix form, we can get Equation 5.

### A.3 PROOF OF LEMMA 1

The Dropout form is trivial. We consider the RotationOut equation. Denote $\boldsymbol{x}^i$ as the $i^{\text{th}}$ term of $\boldsymbol{x}$. The probability distribution of each element of $\widetilde{\boldsymbol{x}}_{\text{Rot}}$ is:

$$\forall j \neq i, \mathbb{P}(\widetilde{\boldsymbol{x}}^i_{\text{Rot}} = \boldsymbol{x}^i + \tan\theta\boldsymbol{x}^j) = \mathbb{P}(\widetilde{\boldsymbol{x}}^i_{\text{Rot}} = \boldsymbol{x}^i - \tan\theta\boldsymbol{x}^j) = \frac{1}{2(D-1)} \tag{20}$$

The joint distribution of each two elements of $\widetilde{\boldsymbol{x}}_{\text{Rot}}$ is:

$$\begin{aligned}
\forall m \neq i, n \neq j, m \neq n : &\mathbb{P}(\widetilde{\boldsymbol{x}}^i_{\text{Rot}} = \boldsymbol{x}^i + \tan\theta\boldsymbol{x}^m, \widetilde{\boldsymbol{x}}^j_{\text{Rot}} = \boldsymbol{x}^j + \tan\theta\boldsymbol{x}^n) \\
=&\mathbb{P}(\widetilde{\boldsymbol{x}}^i_{\text{Rot}} = \boldsymbol{x}^i + \tan\theta\boldsymbol{x}^m, \widetilde{\boldsymbol{x}}^j_{\text{Rot}} = \boldsymbol{x}^j - \tan\theta\boldsymbol{x}^n) \\
=&\mathbb{P}(\widetilde{\boldsymbol{x}}^i_{\text{Rot}} = \boldsymbol{x}^i - \tan\theta\boldsymbol{x}^m, \widetilde{\boldsymbol{x}}^j_{\text{Rot}} = \boldsymbol{x}^j - \tan\theta\boldsymbol{x}^n) \\
=&\mathbb{P}(\widetilde{\boldsymbol{x}}^i_{\text{Rot}} = \boldsymbol{x}^i - \tan\theta\boldsymbol{x}^m, \widetilde{\boldsymbol{x}}^j_{\text{Rot}} = \boldsymbol{x}^j + \tan\theta\boldsymbol{x}^n) \\
=&\frac{1}{4(D-1)(D-3)} \\
\forall i \neq j : &\mathbb{P}(\widetilde{\boldsymbol{x}}^i_{\text{Rot}} = \boldsymbol{x}^i + \tan\theta\boldsymbol{x}^j, \widetilde{\boldsymbol{x}}^j_{\text{Rot}} = \boldsymbol{x}^j - \tan\theta\boldsymbol{x}^i) \\
&\mathbb{P}(\widetilde{\boldsymbol{x}}^i_{\text{Rot}} = \boldsymbol{x}^i - \tan\theta\boldsymbol{x}^j, \widetilde{\boldsymbol{x}}^j_{\text{Rot}} = \boldsymbol{x}^j + \tan\theta\boldsymbol{x}^i) \\
=&\frac{1}{2(D-1)}
\end{aligned} \tag{21}$$

So we have:

$$\mathbb{E}\widetilde{\boldsymbol{x}}^i_{\text{Rot}} = 0, \mathbb{E}(\widetilde{\boldsymbol{x}}^i_{\text{Rot}})^2 = \frac{\mathbb{E}_\theta \tan^2\theta}{D-1}\sum_{j\neq i}(\boldsymbol{x}^i)^2, \mathbb{E}\widetilde{\boldsymbol{x}}^i_{\text{Rot}}\widetilde{\boldsymbol{x}}^j_{\text{Rot}} = -\frac{\mathbb{E}_\theta \tan^2\theta}{D-1}\boldsymbol{x}^i\boldsymbol{x}^j \tag{22}$$

### A.4 DROPOUT BEFORE BATCH NORMALIZATION

Dropout changes the variance of a specific neuron when transferring the network from training to inference. However, BN requires a consistent statistical variance. The variance inconsistency (variance shift) in training and inference leads to unstable numerical behaviors and more erroneous predictions when applying Dropout-before BN.

We can easily understand this using Equation 13. If a Dropout layer is applied right before a BN layer. In training time, the BN layer records the diagonal element of $\text{Var}[\widetilde{\boldsymbol{x}}]$ as the running variance and uses them in inference. However, the actul variance in inference should be the diagonal element of $\text{Var}[\boldsymbol{x}]$ which is small than the recorded running variance (train variance). Li et al. (2019) argues:

P1 Instead of using Dropout, a more variance-stable form *Uout* can be used to mitigate the problem:$\widetilde{\boldsymbol{x}}_i = \boldsymbol{x}_i(1 + r_i)$ where $r_i \sim \text{Unif}[-\beta, \beta]$.

P2 Instead of applying Dropout-a (Figure 1), applying Dropout-b can mitigate the problem.

P3 In Dropout-b, let $r$ be the ratio between train variance and test variance. Expanding the input dimension of weight layer $D$ can mitigate the problem: $D \to \infty, r \to 1$.

Figure 1: Two types of Dropout. The weight layer can be convolutional or fully connected layer.

We revisit these propositions and discuss how to mitigate the problem. For Proposition 1, *Uout* is unlikely to mitigate the problem. The *Uout* noise to different neurons are independent, so the variance shift is the only term to reduce co-adaptations in Equation 16. Though *Uout* is variance-stable, it provides less regularization, which is equivalent to Dropout with a higher keep rate.

Proposition 2 and 3 discuss the positions to insert Dropout. Let $\boldsymbol{x}$ be the output from ReLU layer with $\mathbb{E}[\boldsymbol{x}] = c, \text{Var}[\boldsymbol{x}] = \Sigma$ and $\boldsymbol{y}$ be the input of BN layer. The weight layer in Dropout-a and b

are the same with weight $\boldsymbol{W} \in \mathbb{R}^{n \times D}$. During test time, the inputs to BN layers in Dropout-a and b are the same $\boldsymbol{y} = \boldsymbol{W}\boldsymbol{x}$ with variance $\text{Var}[\boldsymbol{y}] = \boldsymbol{W}\Sigma\boldsymbol{W}^{\mathrm{T}}$. During training time, the inputs are different. In Dropout-a, the formulation is $\widetilde{\boldsymbol{y}}_a = \widetilde{\boldsymbol{W}\boldsymbol{x}}$ where $\mathbb{E}[\widetilde{\boldsymbol{W}\boldsymbol{x}}|\boldsymbol{W}\boldsymbol{x}] = \boldsymbol{W}\boldsymbol{x}$. In Dropout-b, the formulation is $\widetilde{\boldsymbol{y}}_b = \boldsymbol{W}\widetilde{\boldsymbol{x}}$ where $\mathbb{E}[\widetilde{\boldsymbol{x}}|\boldsymbol{x}] = \boldsymbol{x}$. So the training variance for the two types are different. Recall Lemma 1, we have:

$$
\begin{aligned}
\text{Var}[\widetilde{\boldsymbol{y}}_a] &= \mathbb{E}\left[\text{Var}[\widetilde{\boldsymbol{W}\boldsymbol{x}}|\boldsymbol{W}\boldsymbol{x}]\right] + \text{Var}[\boldsymbol{W}\boldsymbol{x}] = \boldsymbol{W}\Sigma\boldsymbol{W}^{\mathrm{T}} + \frac{1-p}{p}\text{diag}(\boldsymbol{W}(\Sigma + \boldsymbol{c}\boldsymbol{c}^{\mathrm{T}})\boldsymbol{W}^{\mathrm{T}}) \\
\text{Var}[\widetilde{\boldsymbol{y}}_b] &= \boldsymbol{W}(\mathbb{E}\left[\text{Var}[\widetilde{\boldsymbol{x}}|\boldsymbol{x}]\right] + \text{Var}[\boldsymbol{x}])\boldsymbol{W}^{\mathrm{T}} = \boldsymbol{W}\Sigma\boldsymbol{W}^{\mathrm{T}} + \frac{1-p}{p}\boldsymbol{W}\text{diag}(\Sigma + \boldsymbol{c}\boldsymbol{c}^{\mathrm{T}})\boldsymbol{W}^{\mathrm{T}}
\end{aligned}
\tag{23}
$$

Let $\boldsymbol{w}$ be $i^{\text{th}}$ row of of $\boldsymbol{W}$ and assume $\boldsymbol{w}_i$ is uniformly distributed on the unit ball. Since the length of $\boldsymbol{w}$ expands the training and testing variance with the same proportion, it does not affect the ratio between training and testing variance, and we can assume the length of $\boldsymbol{w}$ is fixed. The $i^{\text{th}}$ element of actual testing variance is $\boldsymbol{w}\Sigma\boldsymbol{w}^{\mathrm{T}}$. For Dropout-a, the $i^{\text{th}}$ element of running variance (i.e., the training variance) is $\text{Var}[\widetilde{\boldsymbol{y}}_a|\boldsymbol{w}]_i = \boldsymbol{w}\Sigma\boldsymbol{w}^{\mathrm{T}} + \frac{1-p}{p}\boldsymbol{w}(\Sigma + \boldsymbol{c}\boldsymbol{c}^{\mathrm{T}})\boldsymbol{w}^{\mathrm{T}}$. For Dropout-b, the $i^{\text{th}}$ element of running variance is $\text{Var}[\widetilde{\boldsymbol{y}}_b|\boldsymbol{w}]_i = \boldsymbol{w}\Sigma\boldsymbol{w}^{\mathrm{T}} + \frac{1-p}{p}\boldsymbol{w}\text{diag}(\Sigma + \boldsymbol{c}\boldsymbol{c}^{\mathrm{T}})\boldsymbol{w}^{\mathrm{T}}$. Dropout-a and b have the same expected variance shift:

$$
\mathbb{E}_{\boldsymbol{w}}\left[\text{Var}[\widetilde{\boldsymbol{y}}_a|\boldsymbol{w}]_i - \text{Var}[\boldsymbol{y}|\boldsymbol{w}]_i\right] = \mathbb{E}_{\boldsymbol{w}}\left[\text{Var}[\widetilde{\boldsymbol{y}}_b|\boldsymbol{w}]_i - \text{Var}[\boldsymbol{y}|\boldsymbol{w}]_i\right] = \frac{1-p}{p}(\text{trace}(\Sigma) + \boldsymbol{c}^{\mathrm{T}}\boldsymbol{c}) \tag{24}
$$

Though the expected variance shift is the same, the variance of the shift is different. Let $r(\boldsymbol{w})$ be the ratio between the training variance and the testing variance: $r(\boldsymbol{w}) = \text{Var}[\widetilde{\boldsymbol{y}}|\boldsymbol{w}]_i/\text{Var}[\boldsymbol{y}|\boldsymbol{w}]_i$. We have the following observation:

**Observation.** If $\boldsymbol{c} > 0$ which is the case that the activation function is ReLU. The ratio in Dropout-b is more centered: $\text{Var}_{\boldsymbol{w}}[r_b(\boldsymbol{w})] < \text{Var}_{\boldsymbol{w}}[r_a(\boldsymbol{w})] = o(\frac{1}{D})$. Sample $n$ weights to make the weight layer $\boldsymbol{W}$, the maximum ratio in Dropout-a is bigger than the maximum ratio in Dropout-b with high probability: $\max\limits_{1 \leq k \leq n} r_b(\boldsymbol{w_i}) < \max\limits_{1 \leq k \leq n} r_a(\boldsymbol{w_i})$.

According to this observation, Proposition 2 and 3 are basically right but might not be precise. Dropout-b does help mitigate the problem but there might be other reasons. The expected variance shift is the same in Dropout-a and b: $D \to \infty$, $r \nrightarrow 1$. Dropout-b has more stable variance shift among different dimensions. Dropout-a is more likely to have very big training/testing variance ratio, leading to more serious unstable numerical behaviour.

Consider zero-centered Dropout-a in Equation 23: $\text{Var}[\widetilde{\boldsymbol{y}}_a] = \boldsymbol{W}\Sigma\boldsymbol{W}^{\mathrm{T}} + \frac{1-p}{p}\text{diag}(\boldsymbol{W}\Sigma\boldsymbol{W}^{\mathrm{T}})$. The ratio is fixed to be $1/p$ for any weights, i.e. $\text{Var}_{\boldsymbol{w}}[r_a(\boldsymbol{w})] = 0$. It leads to fewer unstable numerical behaviour since there is no extreme variance shift ratio, and we can modify BN layer's validation mode (reduce the running variance by $1/p$ times). Zero-centered Dropout-a can be one solution to mitigate the variance shift problem.

We verified this claim on the CIFAR100 dataset using ResNet110. We apply Dropout between the convolutions of all residual blocks in the third residual stage (18 dropout layers are added). We test three types of Dropout with a keep rate of 0.5: 1) Dropout-a-centered, 2) Dropout-b) and 3) Dropout-b-centered. Following (Li et al., 2019), the experiments are conducted by following three steps: 1) Calculate the running variance of all BN layers in training mode. It is the the training variance. 2) Calculate the running variance of all BN layers in testing mode. It is the the testing variance. Data augmentation and the dataloader are also kept to ensure that every possible detail for calculating neural variances remains exactly the same with training. 3) Obtain $r = \max\{\text{Var}^{\text{train}}/\text{Var}^{\text{test}}, \text{Var}^{\text{test}}/\text{Var}^{\text{train}}\}$ (Note that $\text{Var}^{\text{test}}$ and $\text{Var}^{\text{train}}$ are 64 dimentional vectors). For dropout-a-center, we reduce the running variance by $1/p$ times (We also tried this for the other two dropout, but the results are not better). The obtained ratio $r$ measures the variance shift between training and testing mode. A smaller ratio $r$ is better. The results are averaged over 3 runs and shown in Figure

## A.5 EXPERIMENT IN SPEECH RECOGNITION

We show that our RotationOut can also help train LSTMs. We conduct an Auto2Text experiment on the WSJ (Wall Street Journal) dataset (Paul & Baker, 1992). The dataset is a database with 80 hours

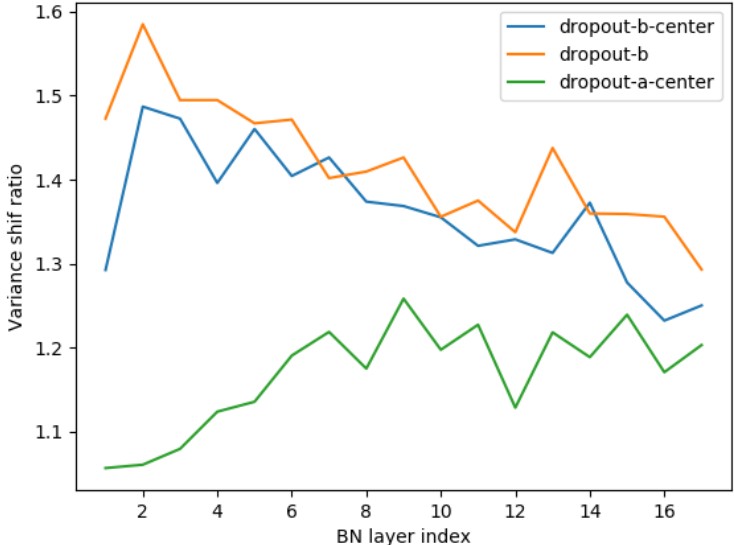

Figure 2: The variance shif ratio for differet dropout variants in the 18 residual blocks from ResNet110, the third residual stage. Lower values are better.

of transcribed speech. The inputs are variable length speech $\boldsymbol{X} \in \mathbb{R}^{T \times L}$ where $T$ is the length and $L$ is the feature dimension for one time step. The labels are character-based words. We use a four-layer bidirectional LSTM network to design a CTC (Connectionist temporal classification) Graves et al. (2006) model. The input dimension, hidden dimension and output dimension of the four-layer bidirectional LSTM network are 40, 512, 137 respectively. We use Adam optimizer with learning rate 1e-3, weight decay 1e-5 and batch size 32, and train the model for 80 epochs and reduce the learning rate by 5x at epoch 40. We report the edit distance between our prediction and ground truth on the "eval92" test set. Table 4 shows the performance of different regularization methods.

Table 4: Auto2Text experiment on the WSJ

| Mothod | Distance |
|---|---|
| No regularization | 9.1 |
| Standard Dropout(kp=0.9) | 8.6 |
| Weight Drop(kp=0.8) | 7.8 |
| Variational Weight Drop(kp=0.8) | 7.5 |
| Locked Drop(kp=0.7) | 7.3 |
| Locked Drop(kp=0.8)+Variational Weight Drop(kp=0.8) | 6.7 |
| RotationOut | 6.8 |
| RotationOut +Variational Weight Drop(kp=0.9) | 6.4 |

## A.6 RETHINKING SMALL BATCHSIZE BATCHNORMALIZATION

BN also introduces noise to the neurons by using the batch mean and variance. The noise to different neurons/channels are independent, so the effect of BN's noise is similar to Dropout. It is widely believed that the noise causes BN performance to decrease with small batch size (Wu & He, 2018; Luo et al., 2018). However, Dropout usually decrease the performance when the keep rate is very low. We study the effect of BN's noise and argue that BN is not a linear operation. The nonlinearity increases when the batch size decreases, which is also one reason for the small batch size BN's performance drop.

Let $\{x_i\}_{i=1}^D$ be one dimension of the data where $D$ is the dataset size. During mini-batch training, one batch of $B$ data $\{x_{b_k}\}_{k=1}^B$ is sampled and the BN operation can be formulate as:

$$\mu_{\mathcal{B}} = \frac{1}{B} \sum_{k=1}^B x_{b_k},$$

$$\sigma_{\mathcal{B}}^2 = \frac{1}{B} \sum_{k=1}^B (x_{b_k} - \mu_{\mathcal{B}})^2, \tag{25}$$

$$\widehat{x}_{b_k} = \frac{x_{b_k} - \mu_{\mathcal{B}}}{\sqrt{\sigma_{\mathcal{B}}^2 + \epsilon}},$$

$$\widehat{y}_{b_k} = \gamma \cdot \widehat{x}_{b_k} + \beta.$$

The batch normalization operation records a running mean $\mu_{\mathcal{B}}$ and running variance $\sigma_{\mathcal{B}}^2$ to be used in testing:

$$E[x] = \mathbb{E}_{\mathcal{B}} \mu_{\mathcal{B}},$$

$$Var[x] = \frac{B}{B-1} \mathbb{E}_{\mathcal{B}} \sigma_{\mathcal{B}}^2, \tag{26}$$

$$\widehat{x}_{b_k} = \frac{x_{b_k} - E[x]}{\sqrt{Var[x] + \epsilon}},$$

We want to check whether the test mode formula can be a good estimation of the training mode formula. Suppose we have a batch of data $\{x_k\}_{k=1}^B$. Denote:

$$\mu_{\mathcal{B}} = \frac{1}{B} \sum_{k=1}^B x_k, \sigma_{\mathcal{B}}^2 = \frac{1}{B} \sum_{k=1}^B (x_k - \mu_{\mathcal{B}})^2, \mu_{\mathcal{B}-1} = \frac{1}{B-1} \sum_{k=2}^B x_k, \sigma_{\mathcal{B}-1}^2 = \frac{1}{B-1} \sum_{k=2}^B (x_k - \mu_{\mathcal{B}-1})^2$$

We have:

$$\frac{x_1 - \mu_{\mathcal{B}}}{\sigma_{\mathcal{B}}} = \sqrt{\frac{B-1}{B}} \frac{x_1 - \mu_{\mathcal{B}-1}}{\sqrt{\sigma_{\mathcal{B}-1}^2 + \frac{1}{B}(x_1 - \mu_{\mathcal{B}-1})^2}} \tag{27}$$

Note that $\mu_{\mathcal{B}-1}$ and $\sigma_{\mathcal{B}-1}^2$ are independent from $x_1$. So the expected output of any $x$ is:

$$\mathbb{E}[\text{Normalize}(x)] = \mathbb{E}_{\mu_{\mathcal{B}-1}, \sigma_{\mathcal{B}-1}^2} \sqrt{\frac{B-1}{B}} \frac{x - \mu_{\mathcal{B}-1}}{\sqrt{\sigma_{\mathcal{B}-1}^2 + \frac{1}{B}(x - \mu_{\mathcal{B}-1})^2}} \tag{28}$$

Let the function in 28 be $f(x, B)$. Easy to know that it is not a linear function (but BN assumes it should be $y = x$!). Suppose the data follows normal distribution, we can plot $f(x, B)$ by Monte Carlo sampling: Figure 3 shows that BN is not a linear operation. The nonlinearity increases when the batch size decreases. It is another important reason for the small batch size BN's performance drop. To validate our conlusion, we propose cross normalization:

$$\mu_i = \frac{1}{B} \sum_{k \neq i}^B x_{b_k},$$

$$\sigma_i^2 = \frac{1}{B} \sum_{k \neq i}^B (x_{b_k} - \mu_i)^2, \tag{29}$$

$$\widehat{x}_{b_i} = \frac{x_{b_i} - \mu_i}{\sqrt{\sigma_i^2 + \epsilon}},$$

$$\widehat{y}_{b_i} = \gamma \cdot \widehat{x}_{b_i} + \beta.$$

For each data in the batch, cross normalization uses the sample mean and variance except itself to comute its normalization mean and variance. In this case, the expectation of operation on any data is striclty linear in expection, but it uses less data.

We do not intend to propose a better alternative for BN but want to check whether the nonlinearity is an important issue for BN when batch size is small. If cross normalization can outperform BN

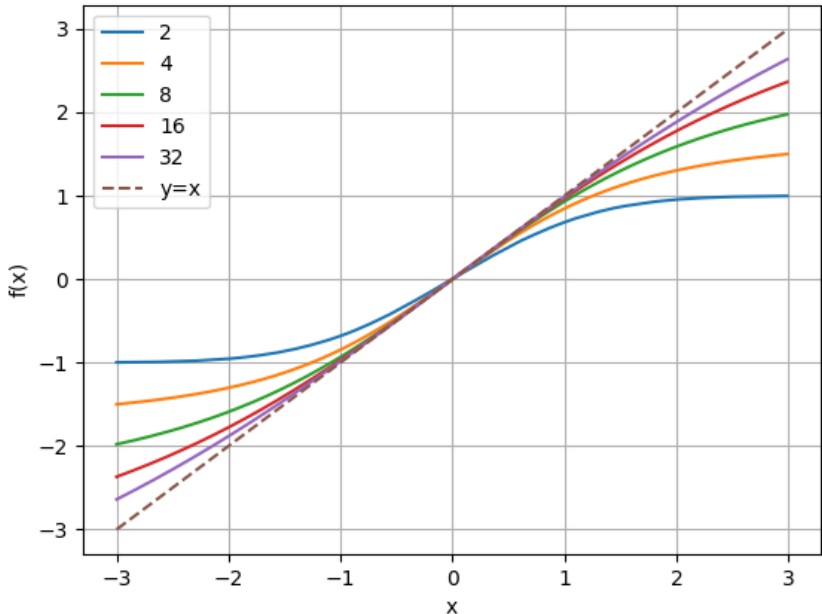

Figure 3: Left: values of $f(B)$ against different values of $B$. Right: the error rates of ImageNet classification against different values of batch size.

in batch size case, then the nonlinearity is definately an important issue. On CIFAR100 dataset, following the settings in our ablation study, ResNet50 with cross normalization has lower test loss when the batch size is 8 and 16. But the test accuracy is almost the same in terms of 95% confidence interval since cross normalization leads to higher variance.

