# OpenReview forum: "RotationOut as a Regularization Method for Neural Network"
_ICLR.cc/2020/Conference — Reject_

### Official Review · AnonReviewer2 · 2019-10-21
**Official Blind Review #2**

**Rating:** 3

**Review:**

This paper proposes a new regularization method to mitigate the overfitting issue of deep neural networks. Specifically, unlike the regular dropout which randomly zeros out some neurons in each layer, the proposed RotationOut rotates the features with a random rotation matrix. The authors argue that the rotational operation can reduces the co-adaptation of features. The experiments have shown some improvement over existing methods.

I have some concerns about the proposed method as follows:

1. In this paper, the authors use the correlation to measure the co-adaptation of features and prove RotationOut can reduce the correlation. However, it is well known that the rotation matrix is an orthogonal matrix, which cannot decorrelate two random variables. So, how can the rotational operation reduce the correlation? It looks a paradox.

2.  Compared with the regular dropout, RotationOut involves more computational overhead. It's better to show the running time of these methods.

3. In Eq.(15), D is the dimension of features. For a large D, such as 224x224 in the imagenet, 1/D is very small. So, the improvement of co_{Rot} over co_{Drop} is very small.

**Experience Assessment:**

I have published in this field for several years.

**Review Assessment: Checking Correctness Of Derivations And Theory:**

I assessed the sensibility of the derivations and theory.

**Review Assessment: Checking Correctness Of Experiments:**

I assessed the sensibility of the experiments.

**Review Assessment: Thoroughness In Paper Reading:**

I read the paper thoroughly.

---

> ### Author Response · Authors · 2019-11-14
> **Response to AnonReviewer2**
>
> Thank you for your time and review. Here are some explanations to your concerns.
>
> 1) The rotation matrix is an orthogonal matrix, which cannot decorrelate two random variables?
>
> Well, it is not the case. Suppose  $x\in\mathbb{R}^d$ follows a multivariate normal distribution in d dimension and $x\sim\mathcal{N}(\mu,\Sigma)$ where $\Sigma$ is the covariance matrix. Suppose the non-diagonal elements of $\Sigma$ is not zero, then the elements of $x$ are correlated to each other. Easy to know that $\Sigma$ is symmetric and semi-positive definite. So the single value decomposition of $\Sigma$ can be written as $\Sigma=USU^{\text{T}}$ where $U$ is an orthogonal matrix where $\Lambda$ is diagonal. Then $U^{\text{T}}x$ follows a multivariate normal distribution in d dimension where the covariance matrix is $\Lambda$. Then the elements of $U^{\text{T}}x$ are independent of each other.
>
> 2) The running time of the proposed method.
>
> We have done a lot to optimize the implementation of RotationOut. Now the runtime is much shorter. Taking  ResNet110 on CIFAR100 for example,  one epoch's training with RotationOut is only 10%~20% slower than that with Dropout. We have released our codes together with our training logs. Feel free to test the performance and runtime speed on your tasks and machines.
>
> 3) For a large D, such as 224x224 in the ImageNet, 1/D is very small. So, the improvement of co_{Rot} over co_{Drop} is very small.
>
> As mentioned in Section 2.3, D refers to the number of filters (channels). So D takes values from 16,32,64 on CIFAR dataset using ResNet110 and and 64,128,256,512 on ImageNet using ResNet50. Also, note that the co-adaption between every two neurons/channels is reduced by p-(1-p)/(D-1) (for dropout, it is reduced by p), but there are D(D-1)/2 pairs. Our definition is just the average of the co-adaption reduction.

---

> > ### Comment · AnonReviewer2 · 2019-11-14
> > **orthonormal matrix is NOT orthogonal matrix**
> >
> > Thanks for your feedback.
> >
> > Please note that U is an orthonormal matrix. It's NOT an orthogonal matrix. They are totally different. Orthogonal matrices preserve the dot product. It definitely cannot reduce the correlation. Thus, I think your paper has a fatal error.

---

> > > ### Author Response · Authors · 2019-11-14
> > > **What is the difference between orthonormal matrix and orthogonal matrix?**
> > >
> > > From my understanding, two vectors $x$ and $y$ are said to be orthogonal is $x^{\text{T}}y=0$ and are said to be orthonormal if we further have $x^{\text{T}}x=1,y^{\text{T}}y=1$.  Then the orthogonal matrix has orthogonal columns or rows and the orthonormal matrix has orthonormal columns or rows. In this case, an orthonormal matrix is an orthogonal matrix.
> > >
> > > We can further give an example of an orthogonal matrix to reduce the correlation. Let $x=(x_1,x_2)^{\text{T}}$ be two random variables. We have $\text{Var}(x_1)= \text{Var}(x_2)=\frac{3}{2}$ and $\text{cov}(x_1,x_2)=-\frac{1}{2}$. Then the coviance matrix of $x=(x_1,x_2)^{\text{T}}$ is $\Sigma=\left[\begin{matrix}
> > > \frac{3}{2},-\frac{1}{2}\\
> > > -\frac{1}{2},\frac{3}{2}
> > > \end{matrix}\right]$ which is symmetric and positive definite. Let $R=\left[\begin{matrix}
> > > \frac{1}{\sqrt{2}},-\frac{1}{\sqrt{2}}\\
> > > \frac{1}{\sqrt{2}},\frac{1}{\sqrt{2}}
> > > \end{matrix}\right]$. We have $RR^{\text{T}}=R^{\text{T}}R=I$ is a rotation matrix. Then the coviance of $Rx$ is $R\Sigma R^{\text{T}}=\left[\begin{matrix}
> > > 2,0\\
> > > 0,1
> > > \end{matrix}\right]$

---

### Official Review · AnonReviewer1 · 2019-10-22
**Official Blind Review #1**

**Rating:** 3

**Review:**

Paper Summary: This paper proposes a novel regularization method for training neural networks. The high-level motivation is to add noise (and thus regularize) neurons in an inter-dependent fashion, unlike existing methods such as DropOut where each neuron is treated independently. The authors evaluate their approach on image and speech classification, and object recognition benchmarks. They also discuss how different regularization schemes might help reduce neuron co-adaptation.

High-level comments: I find the proposed method interesting, and I think the paper is well-written. In particular, I like the exposition of their approach, as well as the common framing of different regularization schemes (in Section 3).

My chief concerns with the paper are:

1. The improvement over state-of-the-art, across tasks, seem marginal and largely is within the reported statistical error margins. From a practical standpoint, I think that alternatives to popular techniques like DropOut should either offer significant benefits empirically, or be computationally more feasible/simpler to implement. RotationOut does not seem to offer major benefits along either of these axes.

2. Even though I find the theoretical comparison of co-adaptation reduction in different regularization methods interesting, its significance is unclear. Based on my understanding, there is no concrete evidence that co-adaptation hampers training or generalization. Moreover, the authors also do not demonstrate that RotationOut actually decreases co-adaptation in a meaningful way---they propose a specific definition for co-adaptation but do not look at this quantity experimentally. There are a number of other proxies for mutual information in the community (such as CCA from arXiv:1806.05759) that the authors should also evaluate.

Given that improvements over prior art are small, I think the paper would have more significance if the authors demonstrate that reducing co-adaptation is important---from the perspective of training/generalization/interpretability and that RotationOut significantly helps with this compared to state-of-the-art approaches.

Other comments:

i. The authors mention that they fix rotate all feature vectors with the same direction but different angles: is the performance actually worse if the directions are also randomized?

ii. In general, I feel the authors should substantiate the various claims they make in Section 3 with experimental evidence. For instance, the authors mention that zero-centered Dropout-a might be more compatible with BatchNorm, but provide no experimental evidence to support this claim.

iii. How did the authors pick the hyperparameters and learning rate schedules for their experiments? These numbers do not seem standard (for instance the learning rate schedule)---did the authors grid over hyperparameters for each of the approaches? I think this is extremely important in papers which are proposing a new approach to establish a rigorous comparison to baselines.

iv. For several of the experiments, the authors report that they use RotationOut only for a few residual blocks. Do the remaining residual blocks have DropOut, or is no regularization applied to these? In the comparisons between different regularization schemes, are the regularizers applied to the same layers? For instance, in the comparison in Table 2, are all the methods applied only to Res3 and Res4?

v. For the results on COCO, is the same regularization method used even for ImageNet pre-training? For instance, does the “RetinaNet, RotationOut + ImageNet” row apply RotationOut even during the ImageNet pre-training? It would be nice to see the results of using “RetinaNet, keep prob = 0.9, block size = 5 + ImageNet” given that “RetinaNet, keep prob = 0.9, block size = 5” performs the best among prior approaches.

Overall, I find the idea of the paper interesting, I am not convinced of its significance. In particular, I think that the improvement over state-of-the-art is marginal and it is not clear that this method actually reduces co-adaptation between neurons in practice, or more broadly, that co-adaptation is a relevant quantity in deep network training. Thus, I am recommending rejection.


**Experience Assessment:**

I have published one or two papers in this area.

**Review Assessment: Checking Correctness Of Derivations And Theory:**

I assessed the sensibility of the derivations and theory.

**Review Assessment: Checking Correctness Of Experiments:**

I carefully checked the experiments.

**Review Assessment: Thoroughness In Paper Reading:**

I read the paper thoroughly.

---

### Official Review · AnonReviewer3 · 2019-10-23
**Official Blind Review #3**

**Rating:** 3

**Review:**

This work proposes a new type of DropOut layer to regularize neural network training. The basic idea is to rotate the features using a random rotation matrix. The authors use Givens rotations to do this in linear time. The authors analyze their DropOut formulation for linear models and contrast it to Bernoulli DropOut. The further provide a probabilistic analysis of the effect on the co-adaption of their DropOut approach. Experimental results are shown on standard classification datasets, object detection, and speech recognition.

Pros:
+ Interesting analysis that supports the idea. Both the analysis of linear models and the co-adaptation analysis help in building intuition about the method.
+ Extensive experiments: The presented experiments cover different applications and are large-scale (e.g. ImageNet). The experiments indicate that there is a small, but consistent, improvement over the baselines.

Cons:
- Parts of the paper are unclear:

1) Equation 2: This equation mentions that the operator is normalized by the cosine of the rotation angle. While the text briefly mentions this and acknowledges that the resulting operator is not a rotation matrix anymore, this is not further mentioned in the text. It is not clear why this normalization was chosen and what would happen if one chooses a proper rotation. When looking at the actual operator that includes this normalization it seems that the term "rotation" is rather misleading. The proposed approach effectively applies a signed and uniformly scaled permutation matrix to the features and adds the results back onto the features.

2) The analysis of the co-adaption is not completely clear to me. Can you elaborate more on the sentence "We use L_1 distance but not L_2..."? Why is it significant here that the variance is a second moment? Equation (15) states that RotationOut reduces co-adaption by a factor of p - (1-p)/(D-1). If we take the extreme case of only two neurons we have that this factor is negative when p < 0.5. More generally, the factor is negative whenever D < 1/p. What does negative co-adaption mean? What does this tell us about this analysis?

3) Section 3.2 is completely disconnected from the rest of the paper. It examines the interdependence between dropout and batch norm, but I don't see how this specifically connects to the contribution at hand.

- The improvement with respect to other DropOut variants is small. RotationOut incurs a higher computational overhead (even if the "rotation" can be done in linear time) in practice according to the authors.

Summary: The authors introduce their idea together with interesting analysis, however, there are some problems with clarity. The practical gains that are afforded by the approach are small, and DropOut, in general, is less and less used (at least in image processing architectures) thus the impact is questionable.

=== Post rebuttal update ===
I'd like to thank the authors for addressing some of my comment. Similar to the other reviewer, I still believe that the improvement even beyond CIFAR is too small. I thus maintain my rating.

**Experience Assessment:**

I have published one or two papers in this area.

**Review Assessment: Checking Correctness Of Derivations And Theory:**

I assessed the sensibility of the derivations and theory.

**Review Assessment: Checking Correctness Of Experiments:**

I assessed the sensibility of the experiments.

**Review Assessment: Thoroughness In Paper Reading:**

I read the paper at least twice and used my best judgement in assessing the paper.

---

### Decision · Program_Chairs · 2019-12-19

**Decision:**

Reject

**Comment:**

All of the reviewers agree the paper has an interesting idea (using rotations of the representation as regularization). However, the reviewers also agree the empirical gains are too insignificant. While the paper shows results on CIFAR, the reviewers mentioned a few other ways to improve performance, such as more complex and unconstrained datasets. These additional experiments would make the effectiveness of proposed approach more convincing.